# Anticancer Effects of Propolis Extracts Obtained with the Cold Separation Method on PC-3 and DU-145 Prostate Cancer Cell Lines

**DOI:** 10.3390/molecules27238245

**Published:** 2022-11-26

**Authors:** Marek Gogacz, Jerzy Peszke, Dorota Natorska-Chomicka, Anna Makuch-Kocka, Katarzyna Dos Santos Szewczyk

**Affiliations:** 1Department of Gynecology, Medical University of Lublin, 20-090 Lublin, Poland; 2Department of Experimental Biotechnology, Decont LLC, 08-500 Ryki, Poland; 3Department of Toxicology, Faculty of Pharmacy, Medical University of Lublin, 20-090 Lublin, Poland; 4Department of Pharmacology, Medical University of Lublin, 20-093 Lublin, Poland; 5Department of Pharmaceutical Botany, Medical University of Lublin, 20-093 Lublin, Poland

**Keywords:** prostate cancer, propolis, cold separation

## Abstract

Plant extracts are increasingly tested for their biological activity and interactions with neoplastic cells. One of such sources of biologically active substances is propolis. This product has been known for thousands of years and is widely used in alternative, folk medicine. Articles describing its effects on the metabolism and cell signaling pathways of neoplastic cells derived from different organs are also published more and more frequently. The purpose of our study was to evaluate the biological activity of propolis extract produced with the cold separation method into hormone-dependent and hormone-independent prostate cancer cell lines. In our study, the propolis extracts showed at least an inhibitory effect on the growth of PC-3 and DU-145 neoplastic cells. Our results suggest that propolis extracts obtained with the cold separation method may be considered as promising compounds for the production of health-promoting supplements.

## 1. Introduction

Prostate cancer is the second most common solid tumor in men and the fifth leading cause of cancer-related death [1].

Propolis is a natural substance of plant origin that reveals its secrets to scientists every day. The multitude of chemical compounds it contains has no equivalent in other mixtures of natural origin. Additional factors determining its composition are the time of the harvesting of the raw material conducted by bees, the influence of environmental factors, climate, bee species and latitude. 

The chemical composition of propolis varies and depends on the place from which it is obtained. Chemical analyses have shown that it is possible to distinguish compounds present in almost all types and extracts of this substance, regardless of the place of collection. The limitation in this case is their content, in some cases reaching less than 1% of their total amount. The rest of them are currently undergoing research on the isolation and verification of biological activity [2,3]. The main issue is how the active compounds are extracted from the raw material. The standard method for obtaining the extract is based on heating the raw material in 70% ethanol. The extract obtained in this way contains up to 12% dry weight, and the content of the wax ballast is limited. Unfortunately, this method has some limitations resulting in the depletion of active compounds, especially lipophilic ones, in the final product. Moreover, the heating of the raw material has a destructive effect on some of the complex compounds with proven biological activity. For this reason, the application of this method is limited only to the industrial production of extract with depleted composition and limited possibilities for therapeutic use.

The maximum use of the biological potential of propolis is based on the use of methods based on the low-temperature separation of active ingredients and/or selective wax ballast precipitation. These methods, although more expensive and time consuming, make it possible to obtain an extract with an abruptly increased biological activity, including those extracts that show these properties on prokaryotic and eukaryotic cells. This is connected, for example, to the presence of compounds in their composition that are susceptible to decomposition under the influence of physical factors. For this reason, among others, cold-produced extracts are increasingly used in research aimed at obtaining new biologically active compounds and potential drugs [4].

In our research, we attempted to demonstrate the benefits arising from the potential use of propolis extracts obtained with the cold separation method.

## 2. Results and Discussion

Androgen signaling plays a key role in the proper development, proliferation and differentiation of the prostate [5,6]. Androgen signal initiation is mediated by the intracellular androgen receptor (AR), a member of the nuclear hormone receptor superfamily. The binding of androgen to the AR transforms the receptor into a biologically active conformation and initiates its translocation to the nucleus. In the next step, binding to specific response elements in the promoter regions of the target genes begins, resulting in the modulation of gene expression, both positively and negatively [5]. Other reports indicate that the AR-mediated transcription of genes may be induced in response to growth factors, such as epidermal growth factor, insulin-like growth factor-1 (IGF-1) and keratinocyte growth factor [7], or cytokines, such as IL-6, in a ligand-independent manner [8]. 

Standard prostate cancer cell lines used in research are DU-145, PC-3 and LNCaP [6]. PC-3 is characterized by the growth mechanism regardless of the lack of androgens, glucocorticoids or fibroblast growth factors in the environment [7]. The results of some studies indicate, however, that the growth and proliferation of these cells is influenced by the growth factors of the epidermis [8]. PC-3 is widely used for studying biochemical changes in cells, for assessing responses to chemotherapeutic agents and for researching potential drugs. In addition, these lines are also used to study the mechanisms of viral infection in mammalian cells that demonstrate an immune response.

A characteristic feature of the PC-3 line is the low activity of the enzymes 5-α testosterone reductase and acid phosphatase. Moreover, cells of this line do not express PSA (prostate specific antigen). Karyotype analyses have shown that PC-3 is triploid with 62 chromosomes. The structure of the chromosomes is also interesting, as analyses have shown the absence of the Y chromosome. From the morphological point of view, it can be concluded that the PC-3 line shows the features of a poorly differentiated adenocarcinoma. They exhibit features common to neoplastic cells of epithelial origin, such as numerous microvilli, junction syndromes, abnormal nuclei and nucleoli, abnormal mitochondria, annular lamina and lipid bodies. PC-3 cells show a high metastatic potential compared with DU-145 cells, which have a moderate metastatic potential [9]. 

DU-145 is a human prostate cancer cell line [10] that was isolated from CNS metastasis derived from primary prostate adenocarcinoma removed during parieto-occipital craniotomy [11]. DU-145 are hormone insensitive and do not express prostate specific antigen (PSA). DU-145 cells have a moderate metastatic potential compared with PC-3 cells, which have a high metastatic potential [9]. DU-145 cells are positive for the androgen receptor.

In the PC-3 and DU-145 cell lines, proliferation is not dependent on the presence of sex hormones. In normal cells, the absence or a limited amount of compounds that induce a signaling pathway leading to division prevents uncontrolled tissue growth. The situation is different in neoplastic cells, which include PC-3 and DU-145. These cells have in their arsenal a number of mechanisms activating internal signaling pathways and abnormal mechanisms preventing the induction of the proper course of apoptosis. Disturbances in the mechanisms of intracellular activation are correlated with the overexpression of specific anti-apoptotic proteins, Bclxl and Bcl-2 (B-cell leukemia/lymphoma) [4]. These proteins, present in excess amounts, inhibit the signaling of apoptosis induced by the secretion of calcium ions from the estrogen complex and block the release of cytochrome C and AIF (apoptosis inducing factor) into the cytoplasm. This results in the inactivation of caspases 3 and 9 and, in addition, no degradation of the laminin protein.

The above-average presence of IKK (IκB kinase) is observed in PC-3 and DU-145 cells. This protein plays a key role in increasing the activity of nuclear transcription factor NF-κB (nuclear factor κB) [12]. NF-κB, in turn, is one of the key regulatory elements in cell mitosis. It influences, among others, the activity of TRAF-1, TRAF-2 (TNF-R-associated factor), Bcl-2 or IAP (inhibitors of apoptosis protein) proteins. In addition, NF-κB is a key element of the signaling pathway influencing the angiogenesis process and the spread of cancer cells in the body, causing metastasis. It is related, inter alia, to the transcription of IL-8, VEGF and metalloproteinases [8,10].

An increase in Akt kinase activity is also observed in PC-3 and DU-145 cells. This protein is one of the key systems that inhibit apoptosis [10].

The inactivation of the apoptotic pathway in PC-3 and DU-145 may be additionally associated with the inactivation of suppressor factors PTEN, p53, Bin1 and PAR4. The p53 regulatory protein seems to be of key importance here, the lack of which or mutation of which leads to excessive proliferation and hormone resistance [8]. Bin1, in turn, is a protein that directly interacts with factor c-myc. Thus, this protein influences apoptosis independently of caspases through cell contraction, cytoplasm vacuolization and DNA degradation. The reduced expression of this protein in PC-3 and DU-145 promotes the overexpression of the proliferative pathway of these tumors [13].

Growth factor TGF-β1 also affects the course of apoptosis in PC-3 and DU-145 cells. The presence of this protein in cells inhibits the proliferation and growth of neoplastic cells, mainly epithelial cells. The mechanism of the inactivation of proliferation is associated with the modulation of the levels of p15, p21 and p27 cyclin inhibitors. PC-3 and DU-145 avoid apoptosis induced by TGF-β1 as a result of the loss of functional receptors for this factor [14].

Although most human prostate cancer cell lines have been reported to have a limited number of androgen receptors (AR negative) [15,16,17], some research results indicate that the DU-145 and PC-3 cell lines have detectable levels of mRNA AR [18,19,20,21,22]. Moreover, the treatment of DU-145 and PC-3 cells with natural interferon β (IFN), which is a potent inhibitory regulator of cell growth in vitro and in vivo -, can cause an increase in the level of the AR receptor protein [23,24].

Similar to other steroid hormone receptors, the AR is a phosphoprotein [25]. The phosphorylation of the AR protein performs several functions, including both its activation and stabilization [21]. In LNCaP cells, the AR protein is detected as two isoforms with apparent molecular weights of 110 and 112 kDa, where the 112 kDa isoform is a phosphorylated form [26]. AR phosphorylation appears to be regulated in a cell-type specific manner [27]. It is currently unclear whether different functions of the AR protein are associated with different phosphorylation sites or whether the phosphorylation of the AR protein at certain sites is important for certain functions.

A large number of mutations in the AR gene have been identified in clinical samples and prostate cancer cell lines [12,28,29]. Among them, the best-known AR variants are those identified in the LNCaP and MDA PCa 2b cell lines. Importantly, the N-terminal and C-terminal regions of the human AR protein are susceptible to amino acid changes, while no mutations have been reported in the region covering amino acids 299–315 [12,29].

### 2.1. Effect of Propolis Extracts Produced with the Low-Temperature Method on the Activity of PC-3 and DU-145 Cancer Cells and WS1 Normal Cells

To test the potential cytotoxic effect of low-temperature propolis extracts on human prostate cancer cells, PC-3 and DU-145 cells were exposed to serial dilutions (1, 10, 25, 50, 75 and 100 µg/mL) of 75 and 80% propolis extracts for 24 h. Analogous studies were also carried out on a line of physiological cells, WS1. On the thus obtained curve, the IC50 values were determined, i.e., the concentration at which the viability of the test cells decreased to half the viability of the control. For comparison, cells were treated with Cisplatin as a reference compound.

The results of viability and metabolic activity were obtained using the MTT test. This test is one of the most widely used in scientific research to assess cell viability, which reflects its cytotoxic activity. The methodology is based on the ability of the enzyme—succinate dehydrogenase—present in living cells with active mitochondrial metabolism to reduce the water-soluble yellow tetrazolium salt MTT (3- (4,5-dimethyl-2-thiazolyl)—2,5-diphenyltetrazole bromide). The reaction produces insoluble purple formazan that picks up as crystals. The resulting precipitate is then dissolved in DMSO and its concentration determined with a spectrophotometer at a wavelength of 550 nm. The amount of reduced colored MTT corresponds in proportion to the mitochondrial activity of the cell. Dead or damaged cells do not metabolize the tetrazolium salt to formazan at all, or the reduction is very limited. This is reflected in the color intensity of the resulting solution, which is proportional to the number of living cells [30].

In general, both extracts were active in the evaluated prostate cell lines. Propolis extracts used in concentrations of 50, 75 and 100 µg/mL led to a very dynamic decrease in the viability of these cancer cells lines (Figure 1 and Figure 2).

Regardless of the cell line, when 10 µg/mL propolis extract was added to the medium, we observed a slight (approx. 10%) increase in the metabolic activity of cells. This phenomenon can be explained by an increase in the antioxidant potential in cells caused by polyphenols and their derivatives against free radicals, which are a side effect of metabolism. A clear destructive effect on the functioning of mitochondria was observed when extracts in an amount exceeding 50 µg/mL were present in the medium. Increased cell metabolism, in correlation with the inhibitory effect of a number of compounds present in propolis extract on the activity of enzymes catalyzing signaling and metabolic pathways in cells, consequently caused the dysregulation of enzymatic mechanisms and cell entry into apoptosis. As a consequence, we observed the inhibition of the proliferation of the cell lines used for the tests and their death in the long term.

In conclusion, the propolis extracts studied in this research showed at least the inhibitory effect on the growth of PC-3 and DU-145 neoplastic cells.

Cisplatin was significantly toxic to the WS1 cell line at the concentration of 50 µg/mL and at higher concentrations (Figure 3). In turn, its cytotoxic effect on prostate cancer cells was insignificant, which can be explained by the resistance of cells to this drug. The androgen-insensitive DU-145 and PC-3 cells are androgen receptor negative and harbor non-functional p53. p53 acts as a tumor suppressor through the induction of growth arrest and apoptosis. These cells do not exhibit cisplatin-induced cell cycle changes, nor do they activate apoptosis [31,32].

### 2.2. Cell-Death Assessment/Apoptosis Analysis

The effect of the tested propolis extracts on the induction of PC-3 and DU 145 cell death was also evaluated using fluorescence imaging, and propidium iodide and Hoechst 33342 staining. In the experiment, the highest concentration of the tested extracts was used, which in the MTT test showed over 75% cell viability—the concentration of 25 µg/mL (Figure 1 and Figure 2)—and an analogous concentration was also used for cisplatin. Concentrations above 25 µg/mL were discarded as they significantly reduced the proliferation and adhesion of the cells used in the experiment. Fluorescence staining showed that 80% and 75% propolis extracts at a concentration of 25 µg/mL statistically significantly induced necrosis in PC-3 and DU 145 cells (Figure 4c–e and Figure 5c–e). It should be noted that a higher fraction of necrotic cells was observed after adding 75% propolis extract to both PC-3 cells and DU 145 (Figure 4c,e and Figure 5c,e). The use of the tested extracts resulted in a higher fraction of necrotic cells in the case of the DU 145 cell line than in PC-3. Cisplatin at a concentration of 25 µg/mL did not show statistically significant changes in the necrotic cell fraction of the tested cancer lines (Figure 4b,e and Figure 5b,e).

### 2.3. Cell Proliferation Inhibition

The BrdU test was used to evaluate the biological activity of the tested propolis extracts and cisplatin on the proliferation and DNA synthesis of PC-3, DU 145 and WS1 cells (Figure 6). The tested propolis extracts statistically significantly inhibited DNA synthesis of PC-3 and DU 145 cell lines at concentrations above 10 μg/mL compared with untreated cells (control). The inhibitory effect of propolis extracts was significantly higher than that of cisplatin on PC-3 and DU 145 cells, the inhibition of proliferation was dependent on the concentration of the extract tested (the higher the concentration of the extract was, the stronger the inhibition of cancer cell proliferation was). The DU 145 cell line was more resistant to the antiproliferative effect of propolis extracts than the PC-3 cell line, and the inhibition of proliferation at the same concentration of the tested extract was stronger in PC-3 cells (Figure 6A) than in DU 145 cells (Figure 6B). The tested extracts also statistically significantly inhibited the DNA synthesis of fibroblasts (WS1 cell line) (Figure 6C). However, it should be noted that the antiproliferative effect of propolis extracts on WS1 cells was usually lower than that of cisplatin at the same concentration.

### 2.4. Activity of Propolis Extracts

The method that is most common and used on an industrial scale for separating organic compounds from propolis mixture is its heating in 70% ethanol. This method is simple and relatively inexpensive, and it allows one to quickly obtain an extract showing biological activity. However, the drawback of this method is the use of high temperatures, leading to the hydrolysis of certain chemical bonds in active compounds, which in turn is associated with the depletion of the final product in active compounds. Therefore, in order to obtain a broad spectrum of biological activity of the extract, it is necessary to use low-temperature methods.

For the cold separation of active compounds, an extragent with a temperature not exceeding 40 °C is used. Under such conditions, practically all of the compounds present in the raw material are dissolved. Ethanol or mixtures of ethanol with tetrahydrofuran (THF) and/or heptane are used as the extragent. In some cases, diethyl ether can also be used. The crude, filtered solution is cascade-cooled to −8 °C. Under such conditions, the selective precipitation of waxes and resins occurs. After their precipitation, the intermediate is concentrated under reduced pressure to a 50 or 80% solids content. With a properly conducted process, depending on the quality of the raw material, the final stage is a product with a content of wax components not exceeding 2%.

Depending on the source of the propolis, this material is characterized by the variability of its organic components. The temperate climate, thus the vegetation from which bees obtain the raw material for the production of propolis, is characterized by a very wide spectrum of biological activities. This is due to the presence of polyphenol derivatives and their glycosides, phenolic acid derivatives, triterpenes, and sterols and their sugar derivatives, as well as their combinations. Propolis extracts produced via the separation of 70% ethanol are not as chemically rich as those produced with the low-temperature method. In the latter case, we identified, using the GC-MS method, at least 3000 compounds, of which only some (about 600) are described in the literature. Over 80% of the compounds have not been clearly identified so far. The selected compounds identified in the propolis extract produced with the low-temperature method that show biological activity towards androgen receptors are presented in Table 1.

The identified compounds in the propolis extract produced by Decont belong to several classes in terms of activity against human cell receptors, as well as against *Procaryota* cells and fungal cells. A special class of the identified compounds are those showing an affinity for estrogen, androgen and progesterone receptors on some cells [33]. These receptors, when attached to an appropriate hormone, initiate a signaling pathway regulating both the metabolism and mitotic activity of cells. This is especially important in the case of neoplastic cells, where the initiation of the hormone-dependent signaling pathway is responsible for the proliferation of cells and, consequently, for the growth of neoplastic tissue. The blocking of this receptor by some of the substances, including those present in propolis extract, may result in the deactivation of the entire signaling pathway and, consequently, in the limitation of the proliferative possibilities of cells and the inhibition of metabolism.

## 3. Materials and Methods

### 3.1. Reagents

All reagents and solvents used in the experiment were purchased from various commercial suppliers and were of the highest purity available. Thiazolyl blue tetrazolium bromide (MTT) and Cisplatin were obtained from Sigma-Aldrich (Steinheim, Germany), dimethyl sulfoxide (DMSO) from POCH S.A. Avantor Performance Materials, Inc. (Gliwice, Poland) and phosphate-buffered saline (PBS) from Mediatech, Inc. Corning Subsidiary (Manassas, VA, USA).

### 3.2. Propolis Extracts

The raw material for extraction was obtained from various sources located in Poland. A necessary condition qualifying for further processing was the complete absence of substances used in plant protection or from the industry. For this purpose, GC-MS purity tests were performed prior to the extraction process.

Four places were identified in Poland whose raw propolis was characterized by the highest parameters in terms of chemical purity and biological properties.

Weighed, raw material, 5000 g per sample, was placed in glass vessels and supplemented with 100% ethanol (5000 mL). The mixture was left for 15 days at room temperature. After this time, the liquid was separated from the solids using filtration. The obtained filtrates were placed in a freezer for 96 h at −2 °C.

After this time, the precipitation of the wax fractions that had been separated under reduced pressure in a Buchner funnel could be observed. The obtained filtrates were placed back in the freezer for 72 h at −6 °C, and the filtration procedure was repeated. The obtained filtrates were concentrated to a concentration of 75 and 80% of dry matter content. The procedure was carried out under reduced pressure on an evaporator, taking care that the temperature of the extract did not exceed 38 °C.

### 3.3. Cell Cultures

Cancer cell lines PC-3 (catalog no. CRL-1435™) and DU-145 (catalog no. HTB-81™), and WS1 (catalog no. CRL-1502™) normal human skin fibroblast cells were purchased from American Type Culture Collection (ATCC; Manassas, VA, USA). The PC-3 cell line was cultured in F-12K Medium (Kaighn’s Modification of Ham’s F-12 Medium) and the DU-145 and WS1 cell lines in EMEM medium (Eagle’s Minimal Essential Medium) supplemented with 10% FBS (Fetal Bovine Serum) and antibiotics, i.e., 100 U/mL penicillin, 100 μg/mL streptomycin and 2.5 μg/mL amphotericin B (PAN-Biotech GmbH, Aidenbach, Germany). Cells were grown in a humidified incubator at 37 °C and 5% CO_2_ atmosphere, in 75 cm^2^ tissue culture flasks.

### 3.4. Cell Cytotoxicity Assay

The inhibitory effects of both propolis extracts on cell growth were assessed using an MTT assay (European Centre for the Validation of Alternative Methods, Database Service on Alternative Methods to Animal Experimentation). Cell viability was determined with a mitochondria-dependent reaction (reduction in mitochondrial dehydrogenase activity) based on the measurement of formazan production from the MTT salt and was expressed as the percentage of viable control cells. The extracts were dissolved in DMSO and subsequently diluted to the required concentration with the respective cell culture medium. The solutions were prepared ex tempore. The cells at a density of 1 × 10^5^ cells/mL in 96-well plates were exposed to various concentrations (1, 10, 25, 50, 75 and 100 µg/mL) of the tested extracts for 24 h at 37 °C. Following incubation, 200 µL of MTT solution (5 mg/mL) was added to each well of a microplate and incubated for 3 h at 37 °C. Subsequently, the culture medium was removed carefully from each well, and 100 µL of DMSO was added. The absorbance of each well was measured at 550 nm using a Power Wave automated absorbance microplate reader (BioTek Instruments, Inc.). Based on the MTT assay results, the IC_50_ values of the tested extracts were derived from the concentration–response curves. The final concentration of DMSO did not exceed 0.1% *v/v*.

### 3.5. BrdU Assay

In order to assess the influence of the tested propolis extracts on the proliferation of the PC-3, DU 145 and WS1 cell lines, the BrdU test was performed. The BrdU assay enables the quantitative determination of 5′-bromo-2′-deoxy-uridine incorporated into newly synthesized DNA of actively proliferating cells. The PC-3, DU 145 and WS1 cell lines were plated at a density of 5 × 10^4^ cells/mL in 96-well culture plates (NUNC, Roskilde, Denmark) incubated under standard conditions. The next day, the culture medium was removed and the appropriate dilutions of the test extracts in culture medium or culture medium alone (control) were added and incubated for 48 h under standard conditions. After 48 h, cell proliferation was assessed according to the manufacturer’s protocol (Cell Proliferation ELISA BrdU; Roche Diagnostics GmbH, Penzberg, Germany).

### 3.6. Fluorescent Cell Death Analysis

PC-3 and DU 145 cells were seeded at 5 × 10^4^ cells/mL on Lab-Tek Chamber Slide (Nunc, Roskilde, Denmark). Then, after 24 h of incubating the cells under standard conditions, the culture medium was replaced with appropriate dilutions of the test extracts in the culture medium. After 24 h, the effect of propolis extracts on the induction of cell death was investigated with the use of Hoechst 33342 fluorescent dyes and propidium iodide. Visualization was performed using a fluorescence microscope, Olympus CKX53, at a magnification of 10×.

### 3.7. Statistical Analysis

The data included in the graphs are presented as means ± standard deviations (±SDs) and were subjected to statistical analyses using the one-way ANOVA with Tukey’s post hoc test (significance was accepted at *p* < 0.05) (GraphPad v.5.01; GraphPad Software, Inc., La Jolla, CA, USA).

## 4. Conclusions

Using an in vitro model, we observed that propolis extracts obtained with cold separation had significant anti-cancer activity on human prostate cancer cell lines PC-3 and DU-145. Both tested extracts possessed significant anticancer activity. Our results provide the basis for further investigation of propolis extracts and the potential isolation of bioactive compounds responsible for anti-cancer properties. However, further in vivo clinical and animal studies of the tested extracts are required to evaluate their modes of action and potential side effects. It is of crucial importance to obtain an explanation of the mechanisms by which these properties are derived from propolis extracts.

## Figures and Tables

**Figure 1 molecules-27-08245-f001:**
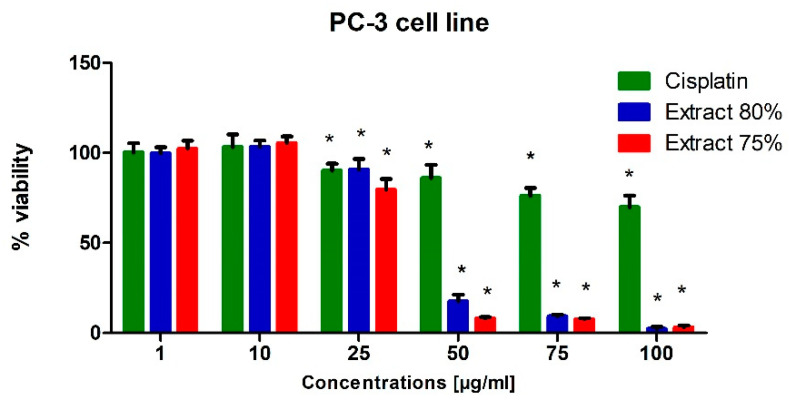
Viability of the PC-3 cell line (%) after 24 h of incubation with various concentrations of extract at 80%, extract at 75% and Cisplatin used as reference compound, measured with MTT test; Significant values (*) compared with Cisplatin with *p* < 0.05 (one-way ANOVA with Tukey’s post hoc test).

**Figure 2 molecules-27-08245-f002:**
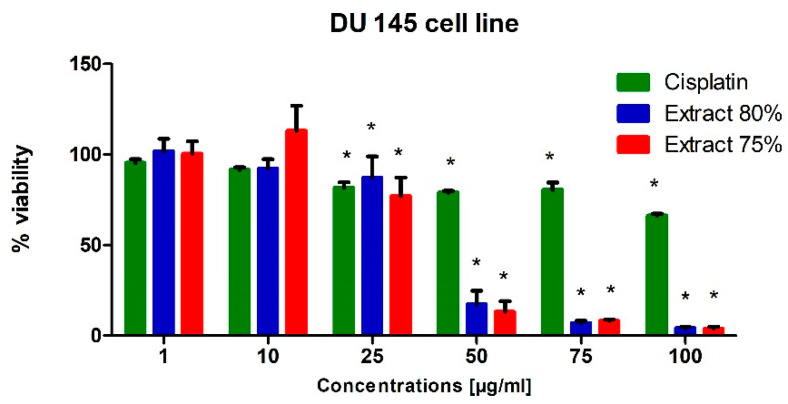
Viability of the DU-145 cell line (%) after 24 h of incubation with various concentrations of extract at 80%, extract at 75% and Cisplatin used as reference compound, measured with MTT test; Significant values (*) compared with Cisplatin with *p* < 0.05 (one-way ANOVA with Tukey’s post hoc test).

**Figure 3 molecules-27-08245-f003:**
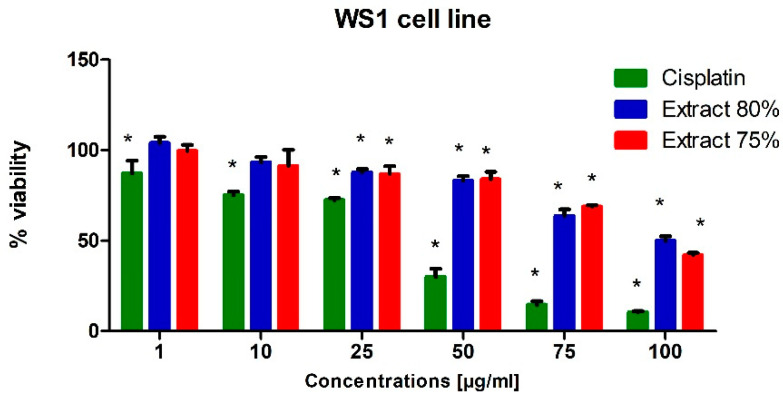
Viability of the WS1 cell line (%) after 24 h of incubation with various concentrations of extract at 80%, extract at 75% and Cisplatin used as reference compound, measured with MTT test; Significant values (*) compared with Cisplatin with *p* < 0.05 (one-way ANOVA with Tukey’s post hoc test).

**Figure 4 molecules-27-08245-f004:**
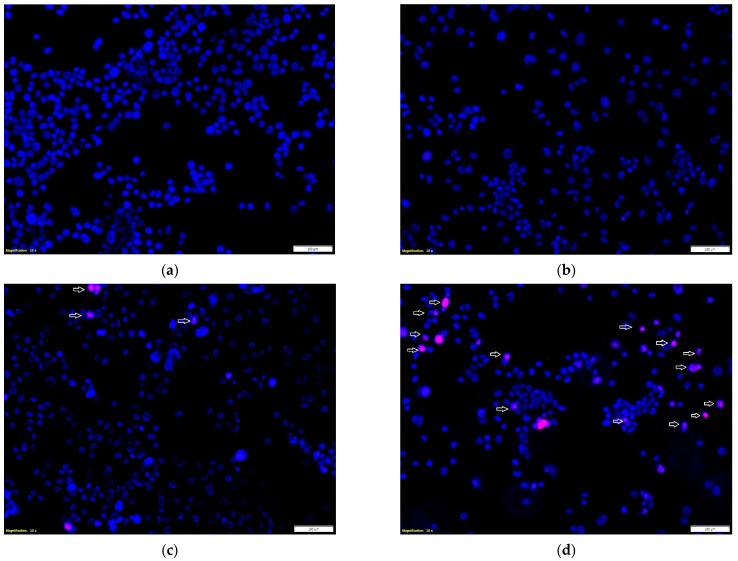
Effects of the tested extracts on the induction of PC-3 cell death: (**a**) control; (**b**) cisplatin at 25 μg/mL; (**c**) 80% extract at 25 μg/mL; (**d**) 75% extract at 25 μg/mL. Propidium iodide (red) and Hoechst 33342 (blue) staining. Magnification, 10×. Necrotic cell nuclei were stained pink. The number of necrotic cells per 100 PC-3 cells was assessed in three visual fields. Data are presented as means ± SDs. Significant values (*) compared with control with *p* < 0.05 (one-way ANOVA with Tukey’s post hoc test) (**e**).

**Figure 5 molecules-27-08245-f005:**
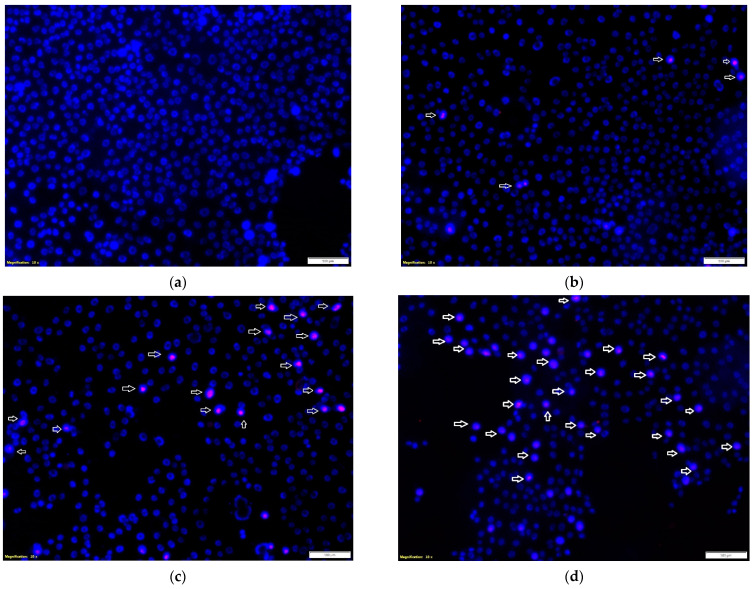
Effects of the tested extracts on the induction of DU-145 cell death: (**a**) control; (**b**) cisplatin at 25 μg/mL; (**c**) 80% extract at 25 μg/mL; (**d**) 75% extract at 25 μg/mL. Propidium iodide (red) and Hoechst 33342 (blue) staining. Magnification, 10×. Necrotic cell nuclei were stained pink. The number of necrotic cells per 100 PC-3 cells was assessed in three visual fields. Data are presented as means ± SDs. Significant values (*) compared with control with *p* < 0.05 (one-way ANOVA with Tukey’s post hoc test) (**e**).

**Figure 6 molecules-27-08245-f006:**
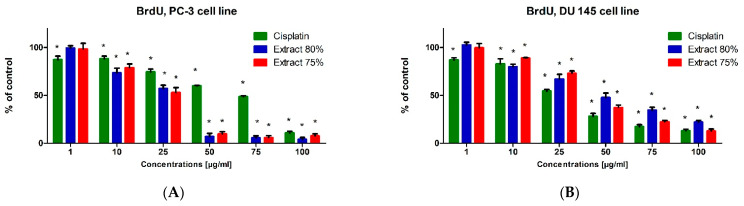
Effect of cisplatin, and 75% and 80% propolis extracts on DNA synthesis of PC-3 (**A**), DU 145 (**B**) and WS1 (**C**) cells, determined using the BrdU test. Results are presented as mean percentage values (% of control) ± SDs (mean value of control = 100%). Values are reported as statistically significant compared with control at * *p* < 0.05 (one-way ANOVA with Tukey’s post hoc test).

**Table 1 molecules-27-08245-t001:** Selected compounds identified in the low-temperature propolis extract that show biological activity towards androgen receptors.

Compound	CAS
(-)-Neoclovene-(I), dihydro-	1000152-82-1
14,17-Nor-3,21-dioxo-β-amyrin, 17,18-didehydro-3-dehydroxy-	1000132-26-8
1H-Cycloprop[e]azulene, decahydro-1,1,7-trimethyl-4-methylene-	72747-25-2
9,19-Cyclolanost-24-en-3-ol, acetate, (3β)-	1259-10-5
Androsterone	53-41-8 (https://commonchemistry.cas.org/detail?cas_rn=53-41-8#_blank accessed on 8 September 2022)
Betulin	473-98-3
Cholest-2-eno[2,3-b]indole, 1′-methyl-5′-methoxy-	1000210-58-3
Dodecanoic acid, ethyl ester	106-33-2
Lup-20(29)-en-3-one	1617-70-5
4-tert-Octylphenol	140-66-9

## Data Availability

Not applicable.

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
