# Peer review of "Anticancer Effects of Propolis Extracts Obtained with the Cold Separation Method on PC-3 and DU-145 Prostate Cancer Cell Lines"

_molecules, 2022, doi:10.3390/molecules27238245_

Round 1
Reviewer 1 Report
Within the paper entitled ''Anticancer Effects of Propolis Extracts Obtained by the Cold Separation Method on PC-3 and DU-145 Prostate Cancer Cell Lines '' the authors, Gogacz et al. have presented a detailed study of obtaining and potential anti-cancer activity of derivatives of of the propolis extracts tested on the different cancer cell lines. These extracts seem to have a quite significant anti-cancer activity. I really appreciated the wide range of the synthetic and analytic methods that have been used within this study. In order to make this study more transparent and clear I would kindly ask authors to elaborate more on the methods they use to calculate error bars they put on their diagrams exposing the anti-cancer activities of their extracts. There is still some more room for an improvement of this decent work of analytical biochemistry. In order to further improve the quality of this study authors should refer to newer works summarizing the really broad spectrum of the anticancer and antioxidant activities of other natural (also some synthetic ones inspired by their natural counterparts) compounds/extracts, and to compare them with the results obtained for their synthesized compounds. The recently published work mentioned under the given citation should be helpful in doing this:https://www.mdpi.com/2218-1989/12/5/451 https://www.mdpi.com/1996-1944/15/7/2544 https://www.mdpi.com/1420-3049/25/22/5243 After implementation of these changes mentioned above will surely make this work more attractive to the broader group of readers.
Reviewer 2 Report
The manuscript by Gogacz et al. describes the study of anticancer activity of propolis extracts obtained by the cold separation method. The results are important and interesting for the journal's readership. However, the manuscript has several drawbacks listed below.
The beginning of Section 2 (lines 57-153) is rather a mini-review than a part of Results and Discussion. This part should be shortened to become more focused on the issues related to the obtained results.
Line 168: The indicated interval from 1 to 550 nm is very strange, as very low wavelengths of about 1 nm are unreachable for UV-VIS spectrophotometers.
Lines 205 and 208: The authors use the term "eluent" while actually they performed extraction of the constituents from propolis. Hence, "extragent" is a more appropriate term.
Lines 205-213: In this paragraph the authors mention tetrahydrofuran, heptane, and diethyl ether as co-extragents with ethanol. However, in Section 3.2 they describe the experimental methodology with the use of ethanol only.
Section 2.2 is titled "Activity of Propolis Extracts. Interaction with Androgen Receptors". However, this section does not contain any description of an interaction with AR, except for the most general phrases. Investigation of such an interaction could imply an affinity study and/or docking computations.
Summarizing, I recommend major revision of the manuscript before acceptance.
Reviewer 3 Report
Dear authors.
The search for new natural compounds with chemopreventive activity is one of the leading directions in food research. Modern life is unfortunately the cause of many diseases and also contributes to an increased risk of cancer. The compounds presented in this paper unfortunately show moderate anticancer activity, and there is no comparison to a reference compound used in the treatment of prostate cancer. Obviously, the activity of natural compounds is different to that of cytostatics, but often combining them with chemotherapy gives promising results. Only one assay was used in the study, which additionally does not measure proliferation. Obviously, the research presented in the paper is valuable, but it is only the beginning of a research study. The pre-publication work requires considerable supplementation, which takes time. Before it is resubmitted for review, it should:
1. expand the scope of the research performed to include assessment of viability, cell growth e.g. SRB, cell cycle, effects on cellular protein expression, apoptosis.
2. the lineage panel should be expanded to include one tumour line and one normal line, preferably epithelial (without this, the activity of the extracts tested cannot be assessed).
3. use a reference compound in the study.
Round 2
Reviewer 2 Report
The revised version of the manuscript has been improved significantly by the authors. In my opinion, the revised version can be accepted for publication.
Reviewer 3 Report
Dear authors.
Thank you for your answers to the questions and for the very solid corrections made.
After the corrections, I have no doubts about the research done and recommend the paper for publication.